# Genomic Characterization of Partial Tandem Duplication Involving the *KMT2A* Gene in Adult Acute Myeloid Leukemia

**DOI:** 10.3390/cancers16091693

**Published:** 2024-04-26

**Authors:** Andrew Seto, Gregory Downs, Olivia King, Shabnam Salehi-Rad, Ana Baptista, Kayu Chin, Sylvie Grenier, Bevoline Nwachukwu, Anne Tierens, Mark D. Minden, Adam C. Smith, José-Mario Capo-Chichi

**Affiliations:** 1Genome Diagnostics & Cancer Cytogenetics Laboratories, Laboratory Medicine Program, University Health Network, Toronto, ON M5G 2C4, Canada; andrew.seto@uhn.ca (A.S.); gregory.downs@uhn.ca (G.D.); olivia.king@uhn.ca (O.K.); shabnam.salehi-rad@uhn.ca (S.S.-R.); ana.baptista@uhn.ca (A.B.); kayu.chin@uhn.ca (K.C.); sylvie.grenier@uhn.ca (S.G.); bev.nwachukwu@uhn.ca (B.N.); 2Department of Laboratory Medicine and Pathobiology, Faculty of Medicine, University of Toronto, Toronto, ON M5S 1A8, Canada; anne.tierens@uhn.ca; 3Division of Hematology and Transfusion Medicine, Laboratory Medicine Program, University Health Network, University of Toronto, Toronto, ON M5G 2C4, Canada; 4Department of Medicine Medical Oncology and Hematology, Princess Margaret Cancer Centre, University of Toronto, Toronto, ON M5G 2M9, Canada; mark.minden@uhn.ca

**Keywords:** acute leukemia, *KMT2A* partial tandem duplication (KMT2A-PTD), structural variation, optical genome mapping (OGM), next-generation sequencing (NGS), multiplex-ligation probe amplification (MLPA)

## Abstract

**Simple Summary:**

Genetic rearrangements of the *KMT2A* gene are associated with diagnostic and prognostic outcomes in the context of myeloid neoplasms. While cytogenetically visible *KMT2A* rearrangements (e.g., translocations) are relatively straightforward to detect by conventional cytogenetics, KMT2A partial tandem duplications (KMT2A-PTD) are too small to be detected by karyotype or FISH. Our study compares the detection of the KMT2A-PTD using three technologies: next-generation sequencing, multiplex-ligation probe amplification, and optical genome mapping.

**Abstract:**

Background: Gene rearrangements affecting *KMT2A* are frequent in acute myeloid leukemia (AML) and are often associated with a poor prognosis. *KMT2A* gene fusions are often detected by chromosome banding analysis and confirmed by fluorescence in situ hybridization. However, small intragenic insertions, termed *KMT2A* partial tandem duplication (KMT2A-PTD), are particularly challenging to detect using standard molecular and cytogenetic approaches. Methods: We have validated the use of a custom hybrid-capture-based next-generation sequencing (NGS) panel for comprehensive profiling of AML patients seen at our institution. This NGS panel targets the entire consensus coding DNA sequence of *KMT2A*. To deduce the presence of a KMT2A-PTD, we used the relative ratio of *KMT2A* exons coverage. We sought to corroborate the KMT2A-PTD NGS results using (1) multiplex-ligation probe amplification (MLPA) and (2) optical genome mapping (OGM). Results: We analyzed 932 AML cases and identified 41 individuals harboring a KMT2A-PTD. MLPA, NGS, and OGM confirmed the presence of a KMT2A-PTD in 22 of the cases analyzed where orthogonal testing was possible. The two false-positive KMT2A-PTD calls by NGS could be explained by the presence of cryptic structural variants impacting *KMT2A* and interfering with KMT2A-PTD analysis. OGM revealed the nature of these previously undetected gene rearrangements in *KMT2A*, while MLPA yielded inconclusive results. MLPA analysis for KMT2A-PTD is limited to exon 4, whereas NGS and OGM resolved KMT2A-PTD sizes and copy number levels. Conclusions: KMT2A-PTDs are complex gene rearrangements that cannot be fully ascertained using a single genomic platform. MLPA, NGS panels, and OGM are complementary technologies applied in standard-of-care testing for AML patients. MLPA and NGS panels are designed for targeted copy number analysis; however, our results showed that integration of concurrent genomic alterations is needed for accurate KMT2A-PTD identification. Unbalanced chromosomal rearrangements overlapping with *KMT2A* can interfere with the diagnostic sensitivity and specificity of copy-number-based KMT2A-PTD detection methodologies.

## 1. Introduction

Structural variants (SVs) affecting the Lysine (K)-Specific Methyltransferase 2A (*KMT2A*) gene, formerly known as *MLL* (myeloid/lymphoid or mixed-lineage leukemia), on chromosome band 11q23.3 are recurrently encountered in acute myeloid leukemia (AML) and are often indicative of early relapse and an overall poor prognosis [1,2,3,4,5,6]. These *KMT2A* gene rearrangements can occur in the context of fusion with other gene partners or result from partial tandem duplication within *KMT2A* (i.e., KMT2A-PTD). KMT2A fusions have been reported in 3% of primary pediatric and adult leukemia, as well as 10% of secondary leukemia, occurring following treatment with DNA topoisomerase II inhibitors [7,8,9,10,11,12,13]. KMT2A-PTDs are identified in 3–10% of AML cases, particularly in up to 25% of patients with a concurrent trisomy of chromosome 11 [1,4].

KMT2A-PTDs are small, intragenic, in-frame duplications within the N-terminal end of *KMT2A*. The breakpoints of these cryptic SVs often occur in flanking intronic sequences of exons 2 to 10 and are mediated by Alu elements [14,15,16,17,18]. Typically, KMT2A-PTDs, such as the recurrent duplication spanning exons 1 to 10 of *KMT2A*, occur below the limit of detection of classical cytogenetics karyotyping or FISH techniques. KMT2A-PTDs were traditionally investigated by classical molecular approaches, such as reverse transcriptase PCR (RT-PCR) and Southern blot [6,19,20,21,22,23,24]. However, next-generation sequencing (NGS) panels have become part of the current standard-of-care testing for AML, as they can enable comprehensive molecular profiling and risk prognostication for sequence variants, in addition to identifying KMT2A-PTDs [16,25,26,27,28,29,30]. 

Identification of KMT2A-PTDs is crucial to the clinical management of AML patients; however, the detection of the KMT2A-PTD is not straightforward. First, these cryptic SVs occur below the resolution of G-banding and FISH [24,31,32]. Second, these intragenic duplications exceed the amplification capacity of traditional PCR strategies. For comparison, the canonical *KMT2A* exons 1 to 10 PTDs are hundreds of times larger than FLT3-ITD (i.e., internal tandem duplications in the *FLT3* gene), which are notoriously difficult to amplify using amplicon-based NGS. Third, analyzing the KMT2A-PTD by NGS requires uniform *KMT2A* sequence coverage, which is difficult to achieve using amplicon-based NGS assays. Lastly, it is challenging to detect SVs, such as KMT2A-PTD, from amplicon or hybrid capture based on short-read NGS. Here, we present our experience with KMT2A-PTD detection in 932 adult AML patients using a custom hybrid-capture NGS panel. To validate the use of NGS data for KMT2A-PTD calling, we confirmed the presence of KMT2A-PTD using two orthogonal methodologies for the detection of copy number alterations: multiplex-ligation probe amplification (MLPA) and optical genome mapping (OGM).

## 2. Materials and Methods

### 2.1. Study Cohort

This consisted of 932 patients that were evaluated for a diagnosis of AML at the Princess Margaret Cancer Centre, Toronto, Canada. DNA was extracted from peripheral blood (PB) or bone marrow (BM) samples of the AML cases studied herein. The study was approved by the University Health Network Research Ethics Board.

### 2.2. Conventional Cytogenetics

G-banding analyses were conducted on all cases with analyzable metaphases. Where there was a suspicion of an 11q23.3 chromosomal rearrangement, *KMT2A* break-apart FISH (Abbott Molecular, Intermedico, Markham, ON, Canada) was conducted to investigate the presence of KMT2A rearrangements.

### 2.3. DNA Target Enrichment and Sequencing

NGS was conducted using a custom hybrid-capture NGS panel (heme-NGS) with probes from OGT (Oxford Gene Technology, Kidlington, UK) targeting clinically relevant myeloid gene regions, such as the entire consensus DNA sequence (CCDS) of *KMT2A*. Data analysis used a custom bioinformatics analysis pipeline following GATK best practices for data pre-processing, where reads were aligned to the GRCh37/hg19 human genome reference (Burrows-Wheeler Aligner v 0.7.12), marking duplicating reads (Picard v1.130) and correcting base quality scores (GATK v3.3.0 Base Quality Score Recalibration Algorithm). Variant calling was performed using varscan v2.3.8, and the mean depth of coverage of the *KMT2A* exon interval was calculated using Picard v1.130. The analytical sensitivity of this NGS panel is 3–5% for the detection of small nucleotide variants as well as larger insertion deletion and duplications.

### 2.4. KMT2A-PTD Detection by NGS

Only samples with a depth of *KMT2A* exon coverage >100× were considered for KMT2A-PTD analysis. For KMT2A-PTD detection, we utilized the ratio of the mean depth of coverage for each of the PTD-specific *KMT2A* exons in N-ter (i.e., exons 1 to 10) relative to reference *KMT2A* exons in C-ter (i.e., exons 27 and 36). Using both exons 27 and 36 safeguards against intragenic structural rearrangements in one or the other exon (Figure 1). The KMT2A-PTD detection algorithm was evaluated on over 1000 patients with hematologic malignancies. The lowest VAF for a KMT2A-PTD detected by this panel was 10%.

### 2.5. KMT2A-PTD Detection by MLPA

Where an adequate sample was available, suspicious KMT2A-PTDs identified by NGS were confirmed using the multiplex-ligation-dependent probe amplification (MLPA) P414 probe mix from MRC-Holland (Amsterdam, The Netherlands). Here, KMT2A-PTD calling was derived from a PTD-specific probe in *KMT2A* exon 4 and a reference probe in *KMT2A* exon 36 (Figure 1). Copy number ratios of exon 4 to exon 36 > 1.3 but <1.65 are in keeping with a KMT2A-PTD.

### 2.6. KMT2A-PTD Detection by OGM

Where an adequate sample was available, the presence of a KMT2A-PTD was further investigated using optical genome mapping (OGM). Unlike NGS and MLPA, OGM is an agnostic KMT2A-PTD detection method that does not require the use of gene-specific probes (Figure 1). Ultra-high-molecular-weight DNA was extracted from patient samples and labeled with the Bionano DLS labeling kit. Labeled molecules were run on nanochannel flow cells and imaged by Bionano Saphyr (Bionano Inc., San Diego, CA, USA). Assembly was performed using the Rare Variant Assembly and visualized using Bionano Access (Solve 3.7, Access 1.7). Analysis of SVs was performed, as described by Levy et al., 2024 [33]. OGM was validated on a cohort of over 200 patients with myeloid malignancies harboring various structural variants, including KMT2A-PTD. This validation showed that OGM allows the analysis of larger and more complex structural variants than KMT2A-PTD at VAF as low as 5%. 

## 3. Results

### 3.1. KMT2A Exon z-Score by NGS

Heme-NGS detected DNA point mutations in myeloid genes variants (e.g., *FLT3*, *RUNX1*, *ASXL1*, *DNMT3A*) in addition to KMT2A-PTD (Appendix A). The presence of a KMT2A-PTD was inferred from the coverage of the PTD-specific *KMT2A* exons (i.e., exons 1 to 10) relative to a reference *KMT2A* exon (i.e., exons 27 or 36). The depth of exon coverage can vary based on several factors, such as sample loading quantity, exon length, number of NGS probes, and guanine/cytosine content (Figure 2A). To account for this variability, we first sought to normalize the *KMT2A* exon coverage by applying heme-NGS to a baseline cohort of 100 cases with no detectable SVs on chromosome 11q23.3 by classical cytogenetics. Within the baseline cohort, we determined the relative mean depth of each PTD-specific *KMT2A* exon to the mean depth of a reference *KMT2A* exon. We then derived the corresponding standard deviation and z-score of these exon ratios (Figure 2B) using the following formula: z-score = [(PTD exon mean depth/reference exon mean depth) − baseline_mean_]/baseline ratio_sd_

### 3.2. KMT2A-PTD Detection Threshold by NGS

AML patients often harbor concurrent numerical and/or SVs involving chromosome 11 that can interfere with KMT2A-PTD detection. We applied heme-NGS to a test cohort of 25 samples with no evidence of KMT2A-PTD but harboring numerical and/or SVs involving *KMT2A*. Within this test cohort, the *KMT2A* exon z-scores were kept within two standard deviations (Figure 2C). We thus determined the cases with *KMT2A* exon z-score > 2 to be suspicious for the presence of a KMT2A-PTD (Figure 2D).

### 3.3. KMT2A-PTD Detection by NGS

Intragenic duplications can occur in any gene region. However, unlike internal duplications seen in other myeloid genes (e.g., FLT3-ITD), KMT2A-PTDs often span over multiple exons and are located in the N-ter portion from exons 1 to 10 of KMT2A (Figure 1, Figure 2 and Figure 3). KMT2A-PTDs are hereby defined as intragenic duplications involving solitary (e.g., Figure 3, case 41) or multiple (e.g., Figure 3, case 1) consecutive exons in the N-ter of *KMT2A* (i.e., exons 1 to 10). Suspicious KMT2A-PTD cases found by NGS have a *KMT2A* exon z-score > 2 for multiple consecutive N-ter exons (Figure 2D). KMT2A z-scores were derived from the ratio of each KMT2A-PTD-specific exon (i.e., 1 to 10) over a reference exon (27 or 36). Determining the z-score for both exon 27 and 36 increases the likelihood of a KMT2A-PTD. Typically, we required that the *KMT2A* z-score is >2 for both exons 27 and 36. Exceptionally, samples showing a *KMT2A* z-score < 2 for only one of these reference exons were considered as KMT2A-PTD-positive, provided that one or multiple flanking exons also had a *KMT2A* z-score > 2 for both exon 27 and exon 36 (e.g., Figure 3, cases 2 and 3). In applying this scheme to a cohort of 932 retrospective AML cases seen at our institution, we identified 41 samples suspicious for a KMT2A-PTD.

### 3.4. KMT2A-PTD Detection by MLPA

We sought to confirm the KMT2A-PTD calls made by heme-NGS for 21 patients, for which sufficient DNA could be obtained for gene dosage by MLPA. MLPA also detected the presence of a KMT2A-PTD (i.e., *KMT2A* exon 4 to exon 36 ratio > 1.3) in 17 cases (Figure 3). All these cases had a duplication involving *KMT2A* exon 4. MLPA yielded inconclusive results in 3 individuals (cases 2, 3, and 35), showing a normal dosage (i.e., copy number ratio: 0.99–1.14) of the KMT2A-PTD-specific probe (exon 4), thus making a PTD of *KMT2A* exon 4 unlikely. Instead, cases 2, 3, and 35 showed an apparent deletion (i.e., copy number: 0.51–0.65) of the KMT2A-PTD reference probe (exon 36), suggesting a C-ter *KMT2A* deletion. However, heme-NGS exon coverage across the entire *KMT2A* coding sequence (i.e., exons 1 to 36) did not suggest a deletion involving the C-ter *KMT2A* exons in any of these three samples. In fact, *KMT2A* z-score values obtained for both exons 27 and 36 (Figure 2D and Figure 3) were in keeping with a KMT2A-PTD in all cases (i.e., case 2: PTD of exons 1 to 10, case 3: PTD of exons 1 to 9, and case 35: PTD of exon 2). MLPA did not detect a KMT2A-PTD in case 32, showing a normal dosage for the PTD-specific and reference probes in *KMT2A* exons 4 and 36, respectively (Figure 3).

### 3.5. KMT2A-PTD Detection by OGM

OGM was performed on 15 AML cases for which concurrent heme-NGS results were also available. OGM and heme-NGS supported the presence of a KMT2A-PTD in 11 of the samples analyzed. These findings were also validated by concordant MLPA results obtained in six samples (Figure 3). Discordant calls between OGM and heme-NGS were seen for cases 3 and 35 with suspicious false-positive calls by heme-NGS. In case 3, OGM detected the presence of two SVs overlapping *KMT2A*: (1) a cryptic insertion of approximately 100 kb of the 5′ KMT2A sequence into chromosome 10 within the *MLLT10* gene, resulting in a cryptic *KMT2A::MLLT10 fusion*, and (2) a 0.82 Mb deletion on the presumed derivative chromosome 11, deleting both proximal (5′) and distal (3′) sequences in the *KMT2A* gene region (Figure 4). It should be noted that this deletion was below the resolution of conventional karyotyping and was likely on the derivative 11, as *KMT2A::MLLT10* fusions are known to often come about from multiple rearrangements. In case 35, OGM detected the presence of a translocation involving chromosomes 11 and 19, juxtaposing the genes *KMT2A* and *ELL* (Figure 5). Conventional karyotyping failed for case 35; however, OGM showed a very complex karyotype with multiple copy numbers and SVs. *KMT2A* break-apart FISH also confirmed the presence of a *KMT2A* rearrangement and evidence of higher ploidy. Additionally seen in Figure 5 is the copy number imbalance at both the *KMT2A* and *ELL* breakpoints, indicating an unbalanced translocation. The imbalance of the translocation resulted in loss of 3′ *KMT2A* sequences distal to the translocation breakpoint, which explains both the false-positive NGS KMT2A-PTD call and the inconclusive MLPA result. A discordant call between OGM and MLPA was seen for case 32, where the MLPA KMT2A-PTD-specific probe in *KMT2A* exon 4 could not detect the presence of a KMT2A-PTD. Instead, both OGM and heme-NGS revealed the presence of a KMT2A-PTD in this individual, suggesting a false-negative result by MLPA (Figure 2D and Figure 3).

## 4. Discussion

Detecting SVs by NGS requires both good coverage of all targeted gene regions and uniformity of sequences covered across all areas targeted. However, these requirements are not often met by many commercial NGS heme panels. Here, we used a hybrid-capture-based NGS assay to overcome the challenges with KMT2A-PTD analysis. Relying on uniform *KMT2A* exonic sequence coverage, we utilized the relative coverage of N-ter (i.e., exon 1 to 10) exons over C-ter exons (i.e., exons 27 or 36) of *KMT2A* to deduce the presence of an intragenic duplication in N-ter (Figure 2). Co-occurring SVs on chromosome 11q23.3, where *KMT2A* is located, may interfere with KMT2A-PTD analysis. To account for these KMT2A-PTD ‘look-alikes’, we established a threshold for *KMT2A* exon coverage (i.e., *KMT2A* exon z-score > 2) after normalizing *KMT2A* exon coverage against a cohort of 100 baseline samples as well as a cohort of 25 test samples, as explained above (Figure 2). 

We analyzed 932 AML cases and used this KMT2A-PTD analysis scheme to identify 41 individuals with a KMT2A-PTD (Figure 3). This included patients with multi-exon (e.g., cases 1–32, 40) as well as single-exon KMT2A-PTD (cases 33–39, 41). The heme-NGS findings of a KMT2A-PTD were also concordant with MLPA (*N* = 17/21, 81%) and OGM (*N* = 11/13, 85%) results, where a sample could be obtained for analysis. In total, we confirmed the heme-NGS KMT2A-PTD results to be true-positive calls in 22 samples based on supportive MLPA and/or OGM results. Of interest, none of these 22 samples had another known SV on chromosome 11q23.3, other than the KMT2A-PTD identified. Of the 22 true-positive KMT2A-PTDs detected by NGS, cases 7, 15, and 20 also had trisomy of chromosome 11 (Table 1), indicating that the heme-NGS KMT2A-PTD detection algorithm is able to identify intragenic *KMT2A* duplications in the context of chromosome 11 aneuploidy. Trisomy of chromosome 11 is seen in up to 25% of AML cases with KMT2A-PTD and can interfere with PTD analysis, considering that there is presence of an intragenic gene duplication in the background of an additional copy of chromosome 11. Due to the uniform gene coverage achieved using heme-NGS, chromosome 11 aneuploidy impacted *KMT2A* exons’ coverage at similar levels across the entire coding region. As a result, the relative ratio of the *KMT2A* N-ter (i.e., exon 1 to 10) to C-ter (i.e., exons 27 or 36) did not vary, whether in the presence of gain or loss of chromosome 11 copies. Thus, KMT2A-PTD detection by heme-NGS was not affected by the co-occurring aneuploidies in cases 7, 15, and 20.

Unlike aneuploidies, however, unbalanced gene rearrangements on chromosome 11q23.3 resulted in differences in exon coverage within the *KMT2A* gene with heme-NGS. When such SVs are present, detecting a KMT2A-PTD is more challenging. For instance, a C-terminal deletion combined with an N-ter with a normal copy number appears to be a relative gain of the N-ter region. Calculating the *KMT2A* exon z-score from two reference exons in C-ter safeguards against some intragenic *KMT2A* C-ter deletions, but not all SVs overlapping with *KMT2A*. In this study, heme-NGS detected the presence of a KMT2A-PTD (Figure 2D) in cases 2 (PTD exon 1 to 10), 3 (PTD exon 1 to 9), and 35 (PTD exon 2); however, the concurrent MLPA and OGM results obtained were not in keeping with the presence of an intragenic *KMT2A* duplication. Indeed, MLPA showed a normal copy number for *KMT2A* exon 4 in all three cases (Figure 3). MLPA results instead suggested these individuals had a relative copy number loss of *KMT2A* exon 27, suggesting that a different *KMT2A* SV was causing abnormal results in these three patients. Of note, G-banded karyotyping did not identify any abnormality on chromosome 11q23 in cases 2 and 3 and was unsuccessful for patient 35. However, OGM showed that cases 3 and 35 had other (non-PTD) *KMT2A* rearrangements: a cryptic *KMT2A::MLLT10* gene fusion and a *KMT2A* deletion (case 3, Figure 4), and a *KMT2A*::*ELL* fusion in the context of a complex genome with 3′ *KMT2A* copy number loss (case 35, Figure 5). Sufficient material to conduct additional investigations was not available for case 2; however, similar to cases 3 and 35, there is also a possibility that a *KMT2A* SV led to a false-positive PTD call in this patient.

NGS panels can combine DNA point mutations and limited analysis of SVs (including copy number analysis) on a single platform (e.g., Appendix A) and are, therefore, extremely useful for standard-of-care testing in AML. However, accurate assessment of SVs using NGS panels is challenging due to the targeted nature of those panels, not to mention the challenges with SV detection using short-read sequencing technologies [34]. Alternative KMT2A-PTD detection approaches should be considered in conjunction to NGS, particularly in patients with overlapping SVs on chromosome 11q23.3. Note that some *KMT2A* rearrangements, including the KMT2A-PTD and cryptic insertions, for example, occur below the resolution of karyotyping (Table 1). Therefore, methodologies such as OGM, that detect genome-wide SVs at a higher resolution than karyotyping, are more suitable for KMT2A-PTD investigations. In the absence of OGM, and since FISH is often integrated into the diagnostic workup of AML cases, reflex FISH testing (with a *KMT2A* break-apart probe) of positive KMT2A-PTD calls made by NGS may assist in some cases to rule out potential PTD mimics. For instance, the signal pattern of the *KMT2A* FISH break-apart probes in case 35 is in keeping with a *KMT2A* rearrangement and will thus invalidate the heme-NGS PTD results of a KMT2A-PTD. Conversely, FISH did not show a *KMT2A* rearrangement in cases 6–8, 10, 16–18, 22, 28, 30, 34, 37, and 38. Here, FISH supported the KMT2A-PTD heme-NGS calls in these 14 patients (Table 1, Figure 2 and Figure 3). However, unlike OGM, cryptic gene rearrangements may not be seen by FISH and still yield false-negative results.

*KMT2A* gene rearrangements resulting in fusions with multiple partner genes are more amenable to classical molecular and cytogenetic techniques and thus have been investigated more thoroughly. In comparison, several questions remain unanswered with respect to KMT2A-PTD, beginning with the very definition of a KMT2A-PTD. For instance, it is unclear whether the following should be considered as KMT2A-PTDs: (1) single-exon intragenic duplications in N-ter of *KMT2A*, or (2) intragenic duplications not involving *KMT2A* exons 1 to 10. Currently, it is unclear whether a single-exon KMT2A-PTD should be interpreted as a variant of uncertain significance or clinically actionable alterations in the context of AML. KMT2A-PTD have traditionally been assayed using classical molecular genetics techniques that provide little information on the composition of these variable genetic lesions. As such, it has not been determined if single-exon KMT2A-PTDs are also indicative of a poorer AML outcome, as shown for multi-exon KMT2A-PTDs.

Higher-resolution technologies, such as NGS and OGM, are elucidating the variable composition in size and copy number of intragenic duplications within *KMT2A*. For instance, a *KMT2A* exon z-score > 2 by heme-NGS suggests a duplication (Figure 2). However, *KMT2A* exon z-score values ranged from 2.02 to 8.52, indicating that more than two copies (i.e., duplication) of the “PTD region” were present in some KMT2A-PTD-positive patients (Figure 3). MLPA (i.e., exon triplication ratio: 1.75–2.15) and OGM also showed similar results. Using OGM, KMT2A-PTDs could be visualized directly. OGM provided the most accurate resolution of KMT2A-PTD composition in size and copy numbers, as shown for individuals harboring two (cases 15 and 20, Figure 6A,B), three (case 36, Figure 6C), or four (cases 9 and 32, Figure 6D,E) PTDs or variable length within *KMT2A*. In comparison, heme-NGS did not allow a clear distinction of cases with two or more PTD copies. For instance, case 36 (2.05–2.14) had a lower *KMT2A* z-score than case 15 (3.03–3.09), yet OGM showed that case 36 had three PTD copies, versus two PTD copies for case 15 (Figure 3). PTD copy number estimates by MLPA were not consistent, as shown for case 9 (ratio: 1.62—quadruplication), case 15 (ratio 1.57—duplication), and case 20 (ratio 1.34—duplication). Comparing OGM, NGS, and MLPA for KMT2A-PTD copy number estimation, it is important to consider the impact of the cancer cell fraction on MLPA and NGS calculations. A higher cancer cell fraction (also with possible presence of LOH) may increase the copy number estimate using MLPA or NGS. See Appendix A for myeloid blasts cell percentage as well as other clinical features of KMT2A-PTD cases investigated herein. Since OGM uses long and intact DNA molecules, the size and label pattern for each molecule in the region can be visualized to more accurately determine the composition of the KMT2A-PTD (Figure 6). Recent studies suggest higher levels of the KMT2A-PTD copy number to be positively associated with relapse and risk of disease transformation [35]. However, at this time, the KMT2A-PTD copy number and complexity are not yet taken into consideration in the clinical management of AML patients, as data on large cohorts with high-resolution KMT2A-PTD analysis are currently unavailable.
cancers-16-01693-t001_Table 1Table 1Characteristics of KMT2A-PTD cases identified by the heme-NGS assay. INC: inconclusive; N/T: not tested. For the OGM nomenclature, values in square brackets (e.g., variant allele frequency (VAF) for structural variants and fractional copy number (fCN) for copy number changes) are reported as proportion of the sample, as per the ISCN 2024 recommendations [36].#G-Banding Optical Genome MappingKMT2AFISHKMT2A-PTDNGSMLPAOGM1N/TN/TYesN/TN/T246,XY,−20,+21[8]/46,idem,der(3)inv(3)(p23q27)inv(3)(q?21q26.2)[12]N/TYesINCN/T348,XY,+8,+19[20]ogm[GRCh38] (8)x3[0.96],del(11)(q23.3)(117,817,690_118,650,394)[0.99],t(10;11)(p12.31;q23.3)(21,642,030;118,493,942)MLLT10::KMT2A [0.98],(19)x3[0.66]N/TYesINCNo4N/TN/TYesN/TN/T545,XX,−7[5]/49,XX,+8,+13,+22[1]/46,XX[17]ogm[GRCh38] (8)x3[0.27],ins(11;?)(q23.3;?)(118,470,405_118,479,068)[0.90],(13)x3[0.28]N/TYesYesYes646,XY,del(11)(p15p11.2)[19]/46,XY[1]ogm[GRCh38] del(11)(p14.3p11.2)(24,233,253_45,484,059)[0.36],ins(11;?)(q23.3;?)(118,470,405_118,479,068)[0.85]NoYesYesYes747,XY,del(11)(p15p11.2),+del(11)[13]/48,XY,+11,+13[6]/46,XY[2]NoYesYesN/T8InconclusiveNoYesN/TN/T946,XY[20]ogm[GRCh38] ins(11;?)(q23.3;?)(118,470,405_118,479,068)[0.81]N/TYesYesYes1046,XY,inv(7)(q11.2q22)?c[22]NoYesYesN/T1146,XY,del(7)(q22q32)[17]/46,XY[3]N/TYesYesN/T1246,XY,add(7)(q32)[20]N/TYesN/TN/T1346,XY,add(2)(p13),add(14)(q24)[16]/46,idem,add(7)(q22)[4]N/TYesN/TN/T1446,XY[20]N/TYesYesN/T1547,XY,del(9)(q13q22),+11[10]ogm[GRCh38] del(9)(q21.11q22.31)(67,717,842_92,504,226)[0.90],(11)x3[0.91],ins(11;?)(q23.3;?)(118,470,405_118,479,068)[0.84]N/TYesYesYes1646,XX,del(12)(p12p13)[22]NoYesYesN/T1746,XY[20]NoYesYesN/T18InconclusiveNoYesYesN/T1946,XX[20]ogm[GRCh38] ins(11;?)(q23.3;?)(118,470,405_118,479,068)[0.91]N/TYesYesYes2047,XY,+11[19]/46,XY [1]ogm[GRCh38] (11)x3[0.90],dup(11)(q23.3q23.3)(118,452,293_118,479,068)[0.85]N/TYesYesYes2146,XY[20]N/TYesYesN/T22InconclusiveNoYesYesN/T2346,XX[21]N/TYesYesN/T24ogm[GRCh38] del(4)(q24)(105061991_105450148)x1[0.5], ins(11;?)(q23.3;?)(118,470,405_118,479,068)[0.86]N/TYesN/TYes25N/TN/TYesN/TN/T2646,XX[21]N/TYesN/TN/T2745,XY,−7[9]/46,idem,+mar [9]/46,XY[3]ogm[GRCh38] (7)x1[0.63],ins(11;?)(q23.3;?)(118,470,405_118,479,068)[0.74] N/TYesN/TYes2846,XY[21]NoYesN/TN/T2946,XY,+1,der(1;14)(q10;q10)[15]/46,XY[5]N/TYesYesN/T3046,XX[20]NoYesN/TN/T31N/TN/TYesN/TN/T32ogm[GRCh38] ins(11;?)(q23.3;?)(118,470,101_118,477,357)[0.84]N/TYesYesYes3346,XY[20]N/TYesN/TN/T34InconclusiveNoYesN/TN/T35ogm[GRCh38] (3,4,7,8,11,12,15,19,20)cx, del(5)(q21.3q32)(108917351_146240776)[0.54],t(11;19)(q23.3;p13.11)(118,493,942;18,499,964)KMT2A::ELL [0.54]YesYesINCNo3647,XY,+8[12]/46,XY[11]ogm[GRCh38] (8)x3[0.42],ins(11;?)(q23.3;?)(118,461,867_118,479,068)[0.62]N/TYesN/TYes3746,XY[21]ogm[GRCh38] ins(11;?)(q23.3;?)(118,470,405_118,479,068)[0.87]NoYesN/TYes3846,XY[24]NoYesN/TN/T3946,XX[20]N/TYesN/TN/T4046,XX,der(6)t(1;6)(q12;p23),del(12)(p11.2p13)[4]/46,XX,del(12)(p11.2p13),der(19)t(1;19)(q12;p13)[3]/46,XX[6]N/TYesN/TN/T4144,XY,der(3)add(3)(p22–24),del(5)(q13q33),−7,−8,add(11)(p15),−12,add(12)(p13),add(13)(q10),add(14)(q32),+mar [9]/46,XY[1]NoYesN/TN/T


KMT2A-PTD are intragenic variants that pose challenges to molecular and cytogenetic diagnostic approaches, as they are difficult to detect with NGS panels due to their size but far too small to ascertain by conventional cytogenetic approaches. Here, we used MLPA, NGS, and OGM to explore KMT2A-PTD detection. In our experience, none of these approaches were able to fully characterize the complex nature of KMT2A-PTDs. *KMT2A* SV analysis by MLPA is limited to a single PTD-specific probe in *KMT2A* exon 4 (Figure 1). Indeed, all the KMT2A-PTD-positive patients identified by MLPA had a PTD involving *KMT2A* exon 4. Non-canonical PTD that do not involve *KMT2A* exon 4 will thus fail detection by MLPA. For instance, case 32, with a PTD spanning over KMT2A exons 2 and 3 by heme-NGS (Figure 2D), was not determined to be KMT2A-PTD-positive using MLPA (Figure 3). Inconclusive MLPA results, such as the ones seen in cases 2, 3, and 35, can be attributed to various factors (e.g., DNA quantity or quality) that are not related to KMT2A-PTD. These inconclusive MLPA results do not provide more information on the *KMT2A* genotype. Thus, additional investigations are required to resolve the MLPA findings.

The heme-NGS panel covers the entire *KMT2A* coding region and thus allows the analysis of canonical (e.g., exons 1 to 10) as well as atypical KMT2A-PTD (e.g., single-exon PTD). Here, we focused our analyses on KMT2A-PTD encompassing exons 1 to 10; however, other researchers have detected larger PTD extending further downstream of *KMT2A* exon 10 using NGS [35]. Heme-NGS provided a better exon-level resolution of KMT2A-PTD than MLPA and/or OGM. Indeed, MLPA (single probe in exon 4) and OGM (four informative labels) only rely on a few indicators for KMT2A-PTD analysis (Figure 1). For example, OGM could not determine the exact breakpoints of the KMT2A-PTD in case 32. In fact, OGM estimated the PTD to span over *KMT2A* exons 3 and 4 (Figure 6E), whereas heme-NGS did not indicate a duplication of *KMT2A* exon 4 (Figure 2D). Absence of a KMT2A-PTD involving exon 4, as suggested by heme-NGS, is also in keeping with the false-negative KMT2A-PTD results by MLPA (Figure 3). In general, OGM could not inform on the extent of KMT2A-PTD beyond exon 5 (Figure 6), considering that there are only two informative labels in *KMT2A* between exons 5 and 16 (Figure 1). Therefore, the approximate KMT2A-PTD length is derived from the size of the duplicated segment on OGM. Overall, the estimated KMT2A-PTD sizes obtained by OGM were comparable to the theoretical Hg38 values (Figure 3) and were also in agreement with the corresponding heme-NGS KMT2A-PTDs size estimate.

Heme-NGS identified eight patients with a single-exon KMT2A-PTD. Note that due to a single PTD-specific probe in *KMT2A* exon 4, MLPA will yield false-negative results in seven of these patients (e.g., cases 33–35: exon 2 KMT2A-PTD, cases 36–37: exon 3 KMT2A-PTD, case 39: exon 8 KMT2A-PTD, and case 41: exon 9 KMT2A-PTD). OGM was applied to cases 36 and 37 and determined that these individuals harbor multi-exon KMT2A-PTDs (i.e., case 36: exons 3 to 5 KMT2A-PTD (Figure 6C), and case 37: exons 3 to 4 KMT2A-PTD). Misidentification of multi-exon KMT2A-PTD as a single-exon KMT2A-PTD may impact clinical management of AML. Currently, such single-exon KMT2A-PTDs will be considered of uncertain significance, whereas multi-exon KMT2A-PTD are clinically actionable and often associated with a poorer AML outcome. Note that the *KMT2A* z-score for exon 4 was close to the PTD threshold in case 37, thus corroborating the OGM findings of multi-exon KMT2A-PTDs involving exons 3 to 4 in this patient. As seen with case 37, seemingly single-exon KMT2A-PTDs via heme-NGS may be true multi-exon KMT2A-PTDs. A low cancer fraction and variable depth of *KMT2A* exons’ coverage can mislead KMT2A-PTD detection by NGS from failure to detect one or multiple *KMT2A* exons involved in a PTD. Indeed, similar to cases 36 and 37, lower *KMT2A* z-scores (i.e., z-score < 2) were observed for N-ter exons of several other patients (e.g., cases 2, 10, and 12) and prevented accurate KMT2A-PTD length assessment. Unlike short-read NGS, OGM enables a high-resolution scan of ultra-long DNA molecules and, therefore, provides a more comprehensive interrogation of *KMT2A*. Only OGM was able to clearly distinguish true KMT2A-PTD from “PTD copy number mimics” in a single assay, as evidenced by cases 3 and 35 (Figure 3, Figure 4 and Figure 5). As shown in this study, assay design can have substantial outcomes on analytical sensitivity and specificity, especially for SVs, such as KMT2A-PTD, greater than 1 kb but smaller than cytogenetic resolution (~10 Mb).

## 5. Conclusions

KMT2A-PTDs are intragenic gene rearrangements of clinical importance to the management of myeloid neoplasms. These alterations have been assayed using a wide range of cytogenetics and molecular approaches; however, the biological consequences of the varied sizes and composition of KMT2A-PTDs remain poorly understood. Novel high-throughput DNA sequencing and/or mapping methodologies are further elucidating the complex nature of KMT2A-PTDs. NGS panels are widely established as standard-of-care testing in AML. While myeloid NGS assays have mainly focused on DNA point mutations’ detection, using NGS data to also detect larger SVs (e.g., KMT2A-PTD detection) adds further utility in many disease contexts. Here, we utilized the relative coverage of *KMT2A* exons to derive the presence of a KMT2A-PTD. Although this approach led to the successful identification of patients with KMT2A-PTD, short-read NGS (or other coverage-based copy number approaches) can yield false-positive results (e.g., cases 2, 3, and 35) and thus warrants caution when interpreting. We have identified the presence of confounding SVs on chromosome 11q23.3 to be a key limiting factor to KMT2A-PTD detection by NGS. Additional testing (e.g., *KMT2A* FISH break-apart) can help distinguish true-positive KMT2A-PTD calls from other KMT2A-PTD mimickers. In our experience, other technologies, such as OGM, provide better resolution and identification of SVs on chromosome 11q23.3, including the KMT2A-PTD. OGM and NGS are complementary approaches that can be used to better characterize and interpret KMT2A-PTDs in a clinical setting. Several questions on the clinical consequences of the size and composition of KMT2A-PTDs are left unanswered and need to be addressed on larger cohorts and likely by additional long-range sequencing characterization and with functional assays.

## Figures and Tables

**Figure 1 cancers-16-01693-f001:**
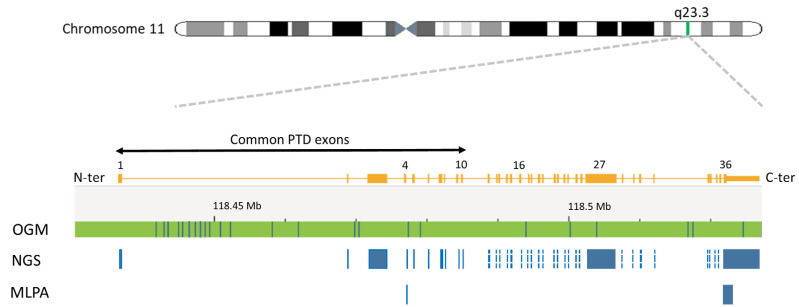
Schematic representation of the 11q23.3 region and the *KMT2A* gene. An ideogram of chromosome 11 is shown (top), and the 11q23.3 chromosome band is indicated. A detailed view of the *KMT2A* gene structure on band 11q23.3 is shown, with exons indicated as rectangles and relevant exons numbered. Below the *KMT2A* gene map, the coverage of each of the techniques used is shown for comparison. The top line shows the chromosome 11 reference map for the *KMT2A* region for optical genome mapping (OGM). Vertical lines on the reference represent a label used to detect structural variants by OGM. DNA is labeled wherever in the genome the sequence CTTAAG is present and usually occurs with an average spacing of approximately 6 kb. As seen by the reference map, the labels are not uniformly spaced. However, it should be noted that the rare variant assembly can detect not only changes in label patterns but also changes in distance between labels, enabling the detection of deletions, duplications, and insertions down to 5 kb. Below the OGM reference, the regions are the *KMT2A* gene regions captured for sequencing by the heme-NGS assay. These include the consensus coding DNA sequences (e.g., exons 1 to 36) of *KMT2A*. KMT2A-PTD analysis using MLPA uses a KMT2A-PTD-specific probe in exon 4, as well as a reference probe in exon 36.

**Figure 2 cancers-16-01693-f002:**
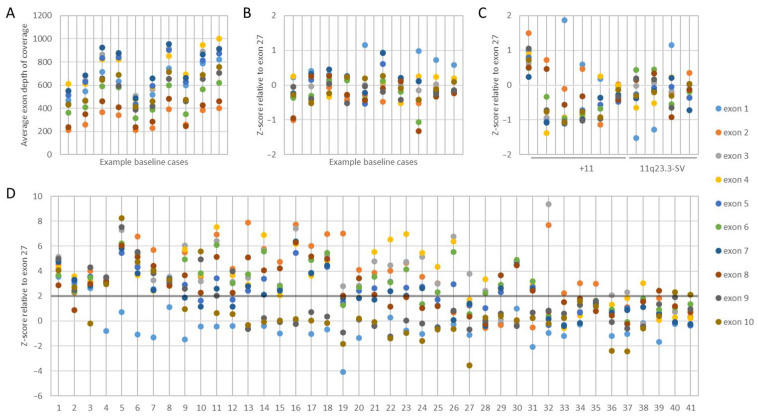
KMT2A-PTD detection using heme-NGS. (**A**) KMT2A-PTD-specific exon 1 to 10 coverage in a cohort of baseline samples with no detectable numerical or structural variants on chromosome 11q23.3. (**B**). *KMT2A* z-score in a cohort of baseline samples, using *KMT2A* exon 27 as a reference exon. (**C**) *KMT2A* z-score in a cohort of patients with numerical (i.e., trisomy 11) or structural variants impacting *KMT2A*, using *KMT2A* exon 27 as a reference exon. (**D**) *KMT2A* z-score ratio in the KMT2A-PTD-positive cases detected by NGS, using *KMT2A* exon 27 as a reference exon.

**Figure 3 cancers-16-01693-f003:**
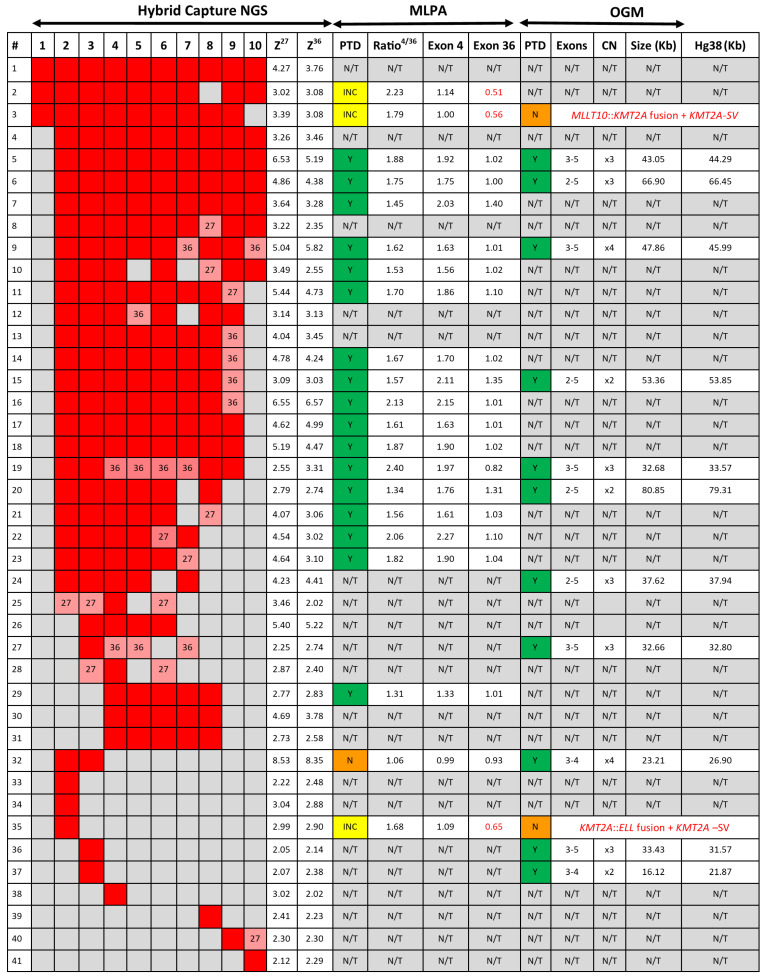
Comparative analyses of NGS, MLPA, and OGM for KMT2A-PTD detection. PTD: partial tandem duplication. Hybrid Capture NGS. Red cells: suspicion of KMT2A-PTD based on KMT2A z-score is >2 for both exons 27 and 36. Pink cells: suspicion of KMT2A-PTD based on KMT2A z-score is >2 for either exons 27 or 36. Grey cells: no evidence of KMT2A-PTD based on KMT2A z-score is <2 for both exons 27 and 36. Z27: Average KMT2A z-score is for exons 27. Z36: Average KMT2A z-score is for exons 36. MLPA: INC: inconclusive results (yellow background); Y (green background): yes—KMT2A-PTD detected; N: no KMT2A-PTD detected (orange background); N/T: not tested; Ratio 4/36: MLPA copy number ratio for the exon 4 KMT2A-PTD-specific over the exon 36 KMT2A reference probe. Ratio below 1 are highlighted in red. OGM. Y (green background): yes—KMT2A-PTD detected; N: no KMT2A-PTD detected (orange background); N/T: not tested. KMT2A-PTD sizes were obtained from the Access analysis software (v 1.7) and compared to approximate sizes derived from the human reference genome Hg38 build.

**Figure 4 cancers-16-01693-f004:**
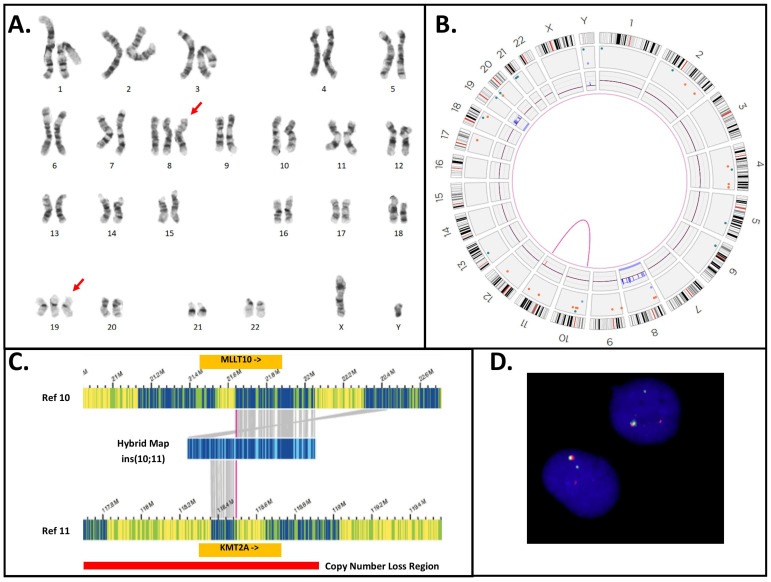
Cytogenetic analyses for Case 3. (**A**) G-banded karyotype, showing a karyotype of 48,XY,+8,+19, extra chromosomes indicated by red arrows. (**B**) Optical genome mapping circos plot, showing trisomy for chromosomes 8 and 19; in addition, a rearrangement between chromosomes 10p and 11q is observed (magenta line). Circos plot elements from the outer ring to the inner ring: chromosome number, ideogram, intra-chromosomal SV (<5 Mb), copy number, aneuploidy bar, intra-chromosomal SV (>5 Mb), or inter-chromosomal SV. (**C**) Genome view of the rearrangement involving chromosomes 10 and 11. Reference chromosome 10 is the top bar and reference chromosome 11 is the bottom bar. A hybrid map (blue bar, middle) shows the alignment of this cryptic insertion to both chromosomes 10 and 11. The match lines (grey) show alignment of labels to the specific segment on the reference sequence, and it can be observed that the sequence from chromosome 11 (*KMT2A* gene) is flanked by the sequence from chromosome 10, indicating that the mechanism of rearrangement is an insertion. (**D**) *KMT2A* break-apart FISH shows a signal pattern consistent with a rearrangement. One set of signals is fused (red + green fusion) and represents the normal chromosome 11. The remaining signals are separated, indicating a rearrangement. Note that the residual separated signals appear diminished compared to the normal chromosome 11 signals—consistent with the deletion of the sequence overlapping the probe region. Abbreviations: Ref: the genome reference pattern for OGM for the specified chromosome.

**Figure 5 cancers-16-01693-f005:**
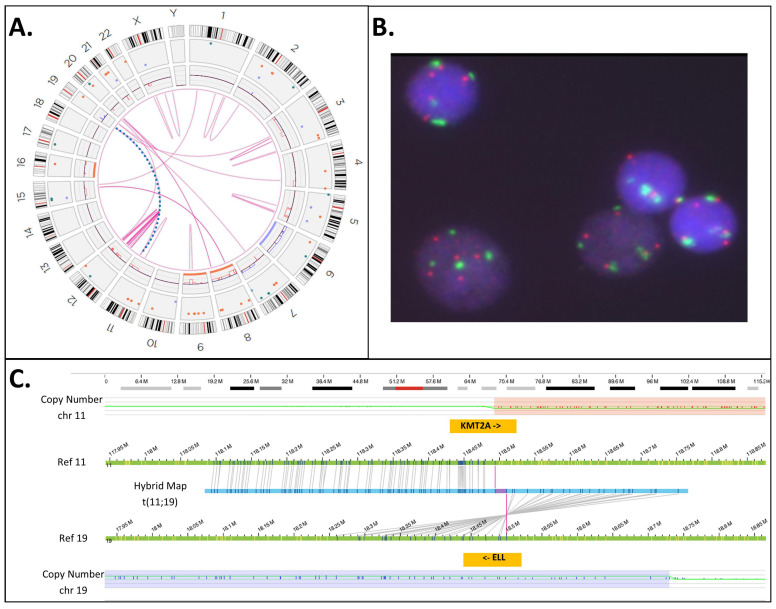
Cytogenetic analyses for Case 35. (**A**) Optical genome mapping circos plot, showing a complex genome with multiple intra- and inter-chromosomal SVs (magenta lines), copy number changes (red and blue boxes on the copy number track of the circos plot), and aneuploidies (orange or blue line spanning the full chromosome). Circos plot elements from the outer ring to the inner ring: chromosome number, ideogram, intra-chromosomal SV (<5 Mb), copy number, aneuploidy bar, intra-chromosomal SV (>5 Mb), or inter-chromosomal SV. A *KMT2A::ELL* translocation was identified and is highlighted among the other inter-chromosomal SVs by a dotted blue line extending from chromosome 11q to chromosome 19p. (**B**) *KMT2A* break-apart FISH shows multiple signals for the 5′ and 3′ probes for *KMT2A*. There are multiple signals for both the 5′ and 3′ probes, indicating a polyploid karyotype. Both fused and separated 5′ and 3′ signals are observed, consistent with a rearrangement of *KMT2A*. (**C**) Genome view of the rearrangement involving chromosomes 11 and 19. Reference chromosome 11 is the top green bar and reference chromosome 19 is the bottom green bar. A hybrid map (blue bar, middle) shows the alignment of this unbalanced translocation to both chromosomes 11 and 19. The match lines (grey) show alignment of labels to the specific segment on the reference genome, and it can be observed that sequence from chromosome 11 in the 5′ *KMT2A* gene region breaks and is joined to the region containing the *ELL* gene on chromosome 19. The copy number track for chromosome 11 appears above the reference (green bar) and shows loss of 3′ *KMT2A* after the breakpoint (light red box). Conversely, the copy number plot for chromosome 19 shows a gain (light blue region), indicating that this rearrangement is unbalanced. Abbreviations: Ref: the genome reference pattern for OGM for the specified chromosome.

**Figure 6 cancers-16-01693-f006:**
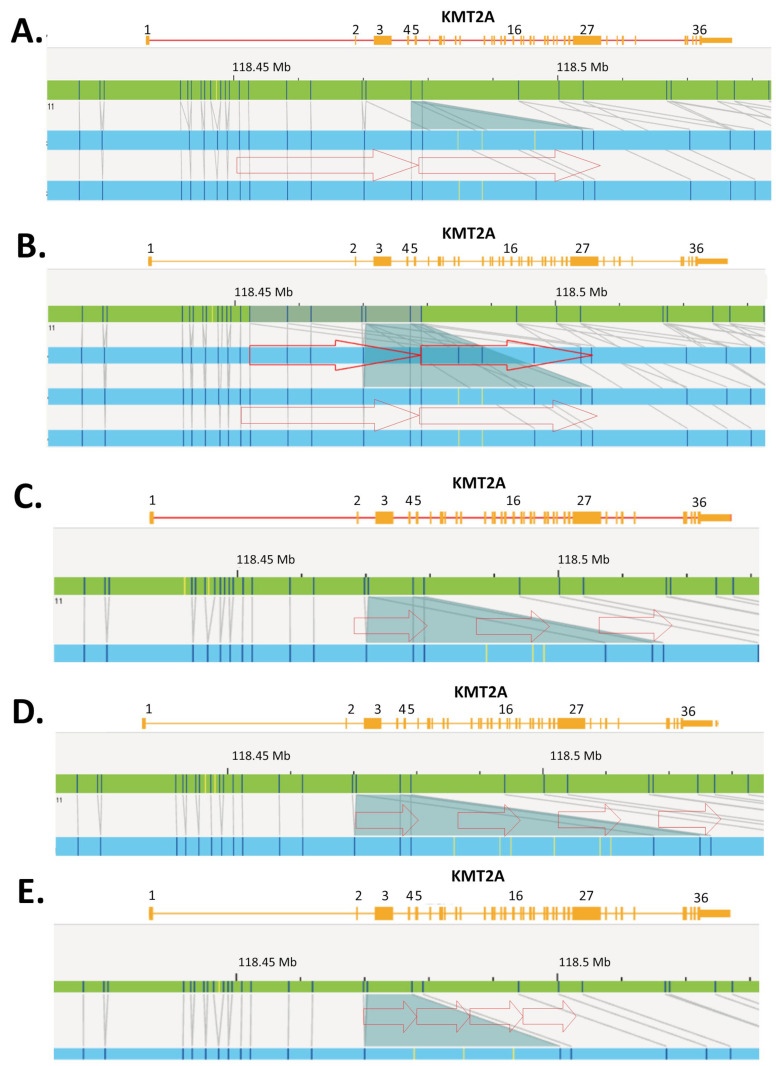
Examples of KMT2A-PTD cases showing size and composition differences detected by heme-NGS and confirmed by OGM. An ideogram showing the *KMT2A* gene region from exons 1 to 36 is depicted. Green bar—reference genome, blue bar—hybrid map, red arrows—duplicated segment. (**A**) Genome view for the KMT2A-PTD in case 15. OGM estimated an SV of 53 kb with two PTD copies encompassing *KMT2A* exons 2 to 5. Note a 5′ PTD breakpoint deep in *KMT2A* intron 1. Note the presence of multiple SV hybrid maps. The top and bottom hybrid maps are highlighted as *KMT2A* intragenic insertions by the Bionano Access software (v1.7). Smaller intragenic duplications (i.e., here, an SV of 27 kb) may be identified as insertions of unknown material, ins(11;?), when insufficient labels are present to definitively align an SV back to the genome reference. (**B**) Genome view for PTD in case 20. OGM estimated an SV of 80 kb with two PTD copies encompassing *KMT2A* exons 2 to 5. Note a 5′ PTD breakpoint deep in *KMT2A* intron 1. Note the difference in KMT2A-PTD size of 27 kb between cases 15 and 20, although the corresponding PTDs in these patients are of approximate lengths. An additional SV hybrid map is listed in this patient. Similar to case 15, the middle and bottom hybrid maps are highlighted as *KMT2A* intragenic insertions. The top hybrid map shows a duplication of overlapping *KMT2A* gene regions. Although not confirmed by OGM, the presence of KMT2A-PTDs in more than one 11q23.3 homologue is likely. Case 20 was shown to harbor a trisomy 11 by OGM and G-banded karyotype. The PTD size difference between cases 15 (i.e., 53 kb) and 20 (i.e., 80 kb), the VAF of the KMT2A-PTDs (>50%), as well as the label patterns and breakpoints for each of the hybrid maps, are additional supportive evidence for KMT2A-PTDs on more than one homologue. (**C**) Genome view for PTD in case 36. OGM estimated an SV of 33 kb with three PTD copies encompassing *KMT2A* exons 2 to 5. The 5′ breakpoint of the PTD was estimated closer to exon 2 than in cases 15 and 20. Note the gaps between the duplicated *KMT2A* segments. It is clear from the spacing between the duplications that the breakpoint is at 3′ of exon 5 due to the amount of sequence without a label; however, the exact breakpoints cannot be determined by OGM. (**D**) Genome view for the PTD in case 9. OGM estimated an SV of 48 kb with four PTD copies encompassing *KMT2A* exons 3 to 5. Note that similar to case 36 above, it is likely that the PTD extended farther than 3′ of *KMT2A* exon 5. The 5′ breakpoint of the PTD was presumed to be even closer than that in cases 15, 20, and 9. (**E**) Genome view for the PTD in case 32. OGM estimated an SV of 23 kb with four PTD copies encompassing *KMT2A* exons 3 to 4. Note that from the label pattern of case 32 (two labels) compared to cases 36 and 9 (three labels), the involvement of *KMT2A* exon 5 in the PTD is unlikely.

## Data Availability

Data are not available due to privacy or ethical restrictions.

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
