# Peer review of "Genomic Characterization of Partial Tandem Duplication Involving the KMT2A Gene in Adult Acute Myeloid Leukemia"

_cancers, 2024, doi:10.3390/cancers16091693_

Round 1
Reviewer 1 Report
Comments and Suggestions for Authors
This study is interesting. In this study, the authors shown that “Genomic characterization of partial tandem duplication involving the KMT2A gene in adult acute myeloid leukemia detected by NGS, MLPA and OGM”. This study provides important information for further understanding of AML The comments about this manuscript are as follows.
1.I suggesting adding some information about the relationship between AML and KMT2A in the section of Abstract.
2. Please define all abbreviations in the text when used for the first time. for example, in page 4 ,line 139,“ GC ”;
3. Please keep the abbreviations consistent, For example, in page 1, line 16,“ KMT2A partial tandem duplications (KMT2A-PTD) ” and in page 1, line 22,“ partial tandem duplication (PTD) ”
4. Studies have shown that the prognosis of AML is related to age, It is necessary to analyse genomic characterization of partial tandem duplication involving the KMT2A gene in young adults and older adults AML.
5. Please provide baseline information about 41 individuals harboring a KMT2A-PTD (age,Initial WBC and so on). In order to compare the potential factors of genomic characterization between different studies.
Author Response
Reviewer #1
We appreciate the feedback received and welcome the opportunity to further improve the quality of this manuscript. We have addressed the comments as per below.
1.I suggesting adding some information about the relationship between AML and KMT2A in the section of Abstract.
We agree with the reviewer that the text in the abstract speaks briefly in regards to the relationship between AML and KMT2A (see below). Note that the abstract as written is already in excess of 300 words and describing more about the relationship and structure of the KMT2A-PTD will make the abstract much longer and unwieldly. Therefore, we believe that the additional detail about KMT2A, partial tandem duplications and AML that is included in the introduction section provides the necessary background for the reader.
“Gene rearrangements affecting KMT2A are frequent in acute myeloid leukemia (AML) and are often associated with a poor prognosis. KMT2A gene fusions are often detected by chromosome banding analysis and confirmed by fluorescence in-situ hybridization. However, small intragenic insertions, termed KMT2A partial tandem duplication (PTD), are particularly challenging to detect using standard molecular and cytogenetic approaches.”
- Please define all abbreviations in the text when used for the first time. for example, in page 4 ,line 139,“ GC ”;
Line 149 (previously line 139) has been edited according to the reviewer’s comment. Guanine and Cytosine are no longer abbreviated as “GC”.
- Please keep the abbreviations consistent, For example, in page 1,line 16, “ KMT2A partial tandem duplications (KMT2A-PTD) ”and in page 1, line 22,“ partial tandem duplication (PTD) ”
Most abbreviations have been converted to KMT2A-PTD as suggested except in cases where the use of KMT2A-PTD would be overly repetitive (e.g. “PTD of KMT2A” instead of “KMT2A-PTD of KMT2A”) or where reference is being made to the duplication of a specific exon.
- Studies have shown that the prognosis of AML is related to age, It is necessary to analyse genomic characterization of partial tandem duplication involving the KMT2A gene in young adults and older adults AML.
The study cohort consists almost exclusively of adult AML patients given the demographics of patients seen at our institution (see Table S1). Other studies have revealed KMT2A-PTD to be an independent prognostic genomic biomarker (e.g. references 1-5, 8-12 of the manuscript) in hematologic cancers including adult and pediatric AML. Age is also often associated with poor disease outcome in various tumour types particularly in hematologic cancers. In the present study, we aimed to focus on the detection of KMT2A-PTD using different methodologies to demonstrate the strengths and limitations of each detection approach and the type of data that is provided. By comparing the approaches, we have also demonstrated how the interpretation of several cases would change depending on the detection technology chosen. The size of the study cohort (i.e. 41 cases) as well as the median follow-up period (i.e. less than 24 months) herein would have prevented optimal analysis of clinical outcomes of KMT2A-PTD as indicated by other studies highlighting the clinical outcome of KMT2A-PTD on larger patient cohorts (e.g. Kim et al. 2019, PMID: 3084064) Follow-up investigations using a comprehensive KMT2A-PTD detection strategy on a larger cohort of patients (i.e. hundred to thousands) will be necessary to address the reviewer’s question. We highlighted this important point in the conclusion section of the manuscript, specifically lines 474 – 494. In summary, several factors prevented the completion of a clinical correlation of KMT2A-PTD with AML outcomes in this study including (1) limited understanding of KMT2A-PTD features (e.g. characterization and interpretation of non-canonical KMT2A-PTD) at the outset of the study, (2) small KMT2A-PTD study cohort (i.e. 41 patients), (3) short follow-up period (i.e. less than 2 years for 2/3 of the study) and (4) homogeneity of sociodemographic parameters (e.g. 2/3 are AML male patients with AML onset> 60 years).
- Please provide baseline information about 41 individuals harboring a KMT2A-PTD (age, Initial WBC and so on). In order to compare the potential factors of genomic characterization between different studies.
Additional clinical and demographic data have been added to supplementary Table 1 (i.e. Table S1).
Reviewer 2 Report
Comments and Suggestions for Authors
The manuscript's examination of KMT2A partial tandem duplications (KMT2A-PTDs) in adult acute myeloid leukemia through three genomic technologies—next-generation sequencing (NGS), multiplex ligation probe amplification (MLPA), and optical genome mapping (OGM)—aims to advance the field's understanding of AML diagnostics. However, its contributions are significantly undermined by methodological inadequacies, a lack of rigorous validation, and an insufficient consideration of clinical utility and outcomes. The reliance on unvalidated analytical thresholds, the limitations in detecting non-canonical PTDs, and the absence of a comprehensive comparison of the technologies' effectiveness contribute to a failure in fully characterizing the complexity of KMT2A-PTDs. These shortcomings underscore the need for a more thorough examination of PTD detection methodologies and their implications for AML prognosis and treatment, suggesting that the study's findings do not markedly advance the current knowledge base in the field. My recommendation on this paper is major revision.
1. The study's reliance on a z-score threshold for NGS-based KMT2A-PTD detection lacks comprehensive validation against known positive and negative controls, raising concerns about the potential for false positives and negatives.
2. MLPA's detection capabilities are hindered by its reliance on a single-probe approach, which may lead to false negatives in PTDs not involving KMT2A exon 4, indicating a critical limitation in its utility for comprehensive PTD detection.
3. The manuscript does not adequately address the resolution limits of OGM, potentially overlooking smaller PTDs crucial for clinical diagnosis and prognosis, thus questioning the method's comprehensive applicability.
4. A significant gap in the manuscript is the lack of consensus and discussion on the clinical significance of KMT2A-PTDs, particularly in differentiating between single and multi-exon duplications and their implications for patient management.
5. The study fails to robustly correlate the findings of KMT2A-PTD sizes and compositions with clinical outcomes, missing an opportunity to enhance understanding of the prognostic significance of KMT2A-PTDs in AML.
Comments on the Quality of English LanguageMinor editing of English language required.
Author Response
Reviewer #2
We appreciate the feedback received and welcome the opportunity to further improve the quality of this manuscript. We have addressed the comments as per below.
Comments and Suggestions for Authors
The manuscript's examination of KMT2A partial tandem duplications (KMT2A-PTDs) in adult acute myeloid leukemia through three genomic technologies—next-generation sequencing (NGS), multiplex ligation probe amplification (MLPA), and optical genome mapping (OGM)—aims to advance the field's understanding of AML diagnostics. However, its contributions are significantly undermined by methodological inadequacies, a lack of rigorous validation, and an insufficient consideration of clinical utility and outcomes. The reliance on unvalidated analytical thresholds, the limitations in detecting non-canonical PTDs, and the absence of a comprehensive comparison of the technologies' effectiveness contribute to a failure in fully characterizing the complexity of KMT2A-PTDs. These shortcomings underscore the need for a more thorough examination of PTD detection methodologies and their implications for AML prognosis and treatment, suggesting that the study's findings do not markedly advance the current knowledge base in the field. My recommendation on this paper is major revision.
- Methodological inadequacies, a lack of rigorous validation:
KMT2A-PTD assessment was largely limited until the recent clinical implementation of novel technologies including Next-Generation Sequencing and Optical Genome Mapping (OGM). Indeed, conventional, orthogonal methodologies such as qPCR or Southern blot provided limited information on important parameters such as KMT2A-PTD size, location and composition (e.g. number of tandem repeats). This manuscript contributes to the current knowledge on KMT2A-PTD investigation by highlighting the technical and analytical challenges of three clinical diagnostic tools with different approaches to the detection of the KMT2A-PTD.
Prior to their implementation, heme-NGS and OGM underwent a thorough validation (e.g. assessment of assay accuracy, precision, robustness, analytical sensitivity, specificity, intra and inter run reproducibility) required for accreditation (e.g. CAP/CLIA).
The MLPA probes used for KMT2A-PTD investigations were available commercially and given their design, these probes will only allow the detection of a KMT2A-PTD encompassing KMT2A exon 4. Note that such an MLPA KMT2A-PTD strategy is widely used by clinical labs. Although not applied clinically at our institution, these KMT2A MLPA probe-mix underwent a thorough validation as done for other MLPA probe-mix or other clinical assays implemented at our institution.
The NGS panel used in this study was developed and validated internally using a similar KMT2A exon coverage scheme for PTD detection similar to the ones described by Conte et al. 2013, PMID: 23702683, McKerrell et al. 2016, PMID: 27389053, Sun et al. 2017, PMID: 27389053. This panel targets the entire consensus coding sequence of KMT2A and thus would allow the detection of both canonical (e.g. PTD involving exons 1-10) and non-canonical KMT2A-PTD provided that they involve an exonic gene region. Note that our conception of a canonical KMT2A-PTD itself is limited by the use of traditional targeted approaches. Note that this test was also assayed for the detection of KMT2A-PTD at an analytical sensitivity of variant allele fraction (i.e. VAF) of 10%. Note that this NGS panel follows CAP/CLIA regulations in keeping with our current certification as a CAP accredited laboratory.
OGM enables a comprehensive analysis of KMT2A-PTD and compared to the NGS strategy deployed, OGM covers both the intronic and exonic regions of KMT2A. Note that compared to MLPA and NGS, high-resolution analysis of all structural variants is accomplished by using OGM. Optical Genome Mapping was rigorously validated in our laboratory for clinical use on a cohort of over 200 patients with a diagnostic myeloid malignancy in patients with various structural variants including KMT2A-PTD and other rearrangements. This validation showed that OGM allows the analysis of larger and more complex structural variants than KMT2A-PTD at VAF as low as 5%. The OGM validation conducted was extremely robust and used a similar validation protocol to Sahajpal et al. 2022, PMID: 36265723.
Note that the OGM validation process is also highlighted in the recent OGM clinical implementation framework (i.e. Levy et al. 2024, PMID: 38164980).Lines 101-103, 109-111, 126-129 were edited to further describe the methodological approach used herein.
Insufficient consideration of clinical utility and outcomes
The goal of this manuscript is to highlight technical and analytical changes with KMT2A-PTD detection. Demonstrating the variation of KMT2A-PTD duplication types is an important first step towards performing larger follow-up studies where the clinical outcomes of KMT2A-PTD subtypes can be determined. Please also see our response to reviewer 1 regarding outcomes. Studying the clinicopathologic features of KMT2A-PTD is out of the scope of the study conducted herein.
- The reliance on unvalidated analytical thresholds
The lower limit of detection of MLPA, NGS and OGM were established at 10%, 10% and 5% respectively, for KMT2A-PTD detection.
- The limitations in detecting non-canonical PTDs, and the absence of a comprehensive comparison of the technologies' effectiveness
Due to the prospective nature of this study, sufficient material could not be obtained to perform a three-way validation (i.e. all patients samples tested by MLPA, NGS and OGM) on all the samples processed. Applying MLPA, NGS and OGM in parallel to the 932 patients investigated would be necessary to complete a comprehensive analysis of the effectiveness of MLPA, NGS and OGM for KMT2A-PTD detection. Such a study would require substantial resources and patient material that we did not have. Instead, we sought to confirm positive KMT2A-PTD calls by NGS using MLPA and OGM where sufficient sample could be obtained for cross-validation. NGS appeared as an obvious starting point as NGS panels are applied to standard of care of AML patients. Of crucial importance to a follow-up study on the clinical impact of KMT2A-PTD, NGS also captures DNA point mutations in numerous myeloid genes (e.g. DNMT3A, IDH1, IDH2, FLT3) that are of diagnostic, therapeutic and prognostic importance in AML. As KMT2A-PTD are rarely found in isolation, such biomarkers can interfere with the prognostic value of KMT2A-PTD. Similarly, several genetic lesions that are best identifiable using OGM can also have an impact on the overall prognosis of KMT2A-PTD on AML (e.g. recurrent rearrangements, complex karyotype). Identifying the challenges with KMT2A-PTD analysis using complementary approaches such as MLPA, NGS and OGM is thus even more important and should be addressed. Here, we have identified important limitations of MLPA, NGS and OGM with assessing the presence of KMT2A-PTD and determining KMT2A-PTD size, copy number and complexity. Performing a clinical study on KMT2A-PTD without better characterizing these parameters of would undermine the success of such a study. For example, the concepts of non-canonical or other complex KMT2A-PTD are still recent and have only been brought up to attention using technologies such as OGM and NGS. Both OGM and NGS are limited in their abilities to fully characterize such KMT2A-PTD. Follow-up studies using long range sequencing and/or methylation analysis will help shed more light on the functional significance of non-canonical KMT2A-PTDs, which is beyond the scope of this study.
- The study's reliance on a z-score threshold for NGS-based KMT2A-PTD detection lacks comprehensive validation against known positive and negative controls, raising concerns about the potential for false positives and negatives.
KMT2A-PTD coverage using NGS is often achieved by comparing the relative coverage of KMT2A-PTD exons. The KMT2A-PTD exon ratio scheme was successfully applied for KMT2A-PTD detection as shown by McKerrell et al. 2016. Here, we put forward a similar scheme with an additional exon ratio (i.e. ratio to exon 37) to further improve KMT2A-PTD detection. Similar KMT2A exon coverage approaches have been described for KMT2A-PTD detection using NGS (e.g. Conte et al. 2013, PMID: 23702683, Sun et al. 2017, PMID: 27389053). Our KMT2A-PTD detection approach was thoroughly validated on over 1000 AML patients including cases with confirmed KMT2A-PTDs.
- MLPA's detection capabilities are hindered by its reliance on a single-probe approach, which may lead to false negatives in PTDs not involving KMT2A exon 4, indicating a critical limitation in its utility for comprehensive PTD detection.
We agree with the reviewer on this point. Many labs use MLPA for PTD detection and we felt it was important to highlight the variety of size and composition of KMT2A-PTDs with various detection strategies. Our manuscript illustrates this critical limitation. See specifically Figure 3 (patients 32-38) that do not include exon 4 in the PTD.
- The manuscript does not adequately address the resolution limits of OGM, potentially overlooking smaller PTDs crucial for clinical diagnosis and prognosis, thus questioning the method's comprehensive applicability.
OGM’s resolution limit using the Rare Variant Pipeline has been well published and documented:
Sahajpal et al. 2022, PMID: 36265723
Valkama A et al. 2023, PMID: 36831635.
Using the Rare Variant Assembly insertions/deletions/duplications down to 5kb (and often below) can reliably be detected at variant allele frequencies down to 5%. We have confirmed OGMs ability to do this within our laboratory on many different types of structural variants.
While it is possible that a small partial tandem duplication below the resolution of OGM could occur (<5kb), there is no published evidence that this duplication would have the same prognostic significance of canonical KMT2A-PTDs that are usually ~30kb in size. Our data from both NGS and OGM suggests that the smallest “PTD” in our cohort of 41 patients is approximately 16kb in size, well above the 5kb limit for structural variants detected by OGM. Further, KMT2A-PTDs smaller than this have not been published nor would they be detected by many of the currently used assays in the clinic. Their clinical significance, if any, would still need to be determined.
- A significant gap in the manuscript is the lack of consensus and discussion on the clinical significance of KMT2A-PTDs, particularly in differentiating between single and multi-exon duplications and their implications for patient management.
We agree with the reviewer that there is a current gap in understanding regarding the clinical significance of single and multi-exon KMT2A-PTDs and patient management implication. In fact, we explicitly state this in the conclusion section:
“Several questions on the clinical consequences of the size and composition of KMT2A-PTD are left unanswered and need to be addressed on larger cohorts and likely by additional long-range sequencing characterization and with functional assays.”
The focus of this manuscript is that many current detection methods for KMT2A-PTD in use today (NGS panel copy number estimation, MLPA) are inadequate to properly characterize KMT2A-PTDs. There is clearly variability in size and composition of these structural variants that we are only beginning to appreciate. And as we commented, we believe that a more comprehensive evaluation of KMT2A-PTDs with smaller or larger sizes need to be evaluated both at the structural and functional level to determine their significance.
- The study fails to robustly correlate the findings of KMT2A-PTD sizes and compositions with clinical outcomes, missing an opportunity to enhance understanding of the prognostic significance of KMT2A-PTDs in AML.
As mentioned in response to point #4, considerable work needs to be done to characterize non-canonical KMT2A-PTDs. As we have detected some non-canonical PTDs in the current study our emphasis was to demonstrate that their detection is possible, with technologies such as OGM, and that current strategies like MLPA and NGS miss a great deal of the compositional differences of these PTDs. Individually, the number of non-canonical PTDs we have in this manuscript would be insufficiently statistically powered to derive any robust and reproducible evidence of impact on clinical outcomes. The focus of this manuscript was meant to highlight the challenges of current gold standard clinical practice for KMT2A-PTD detection with newer high resolution cytogenomic methods for structural variant detection and to raise questions for further study. Accurate detection and characterization of KMT2A-PTDs must first occur before any wide-spread assessment of clinical outcomes can be achieved. Overall, several factors prevented the completion of a clinical correlation of KMT2A-PTD with AML outcomes in this study including (1) limited understanding of KMT2A-PTD features (e.g. characterization and interpretation of non-canonical KMT2A-PTD), (2) small KMT2A-PTD study cohort (i.e. 41 patients), (3) short follow-up period (i.e. less than 2 years for 2/3 of the study) and (4) homogeneity of sociodemographic parameters (e.g. 2/3 are AML male patients with AML onset> 60 years).
Reviewer 3 Report
Comments and Suggestions for Authors
The manuscript by Seto and colleagues reports the detection and study of KMT2A-PTD using three different technologies. The manuscript is well written but lack some relevance in term of clinical data. It would be interesting to study the characteristics of patients with KMT2A-PTD compared to other patients. Is there higher genomic instability for these patients ? Is PTD linked to specific oncogenic mutations ?
Comments on the Quality of English LanguageMinor revision.
Author Response
Reviewer #3
We appreciate the feedback received and welcome the opportunity to further improve the quality of this manuscript. We have addressed the comments as per below.
Comments and Suggestions for Authors
The manuscript by Seto and colleagues reports the detection and study of KMT2A-PTD using three different technologies. The manuscript is well written but lack some relevance in term of clinical data. It would be interesting to study the characteristics of patients with KMT2A-PTD compared to other patients. Is there higher genomic instability for these patients ? Is PTD linked to specific oncogenic mutations ?
The complexity of KMT2A-PTD has been brought to attention with the recent application of techniques such as NGS and OGM. Here we sought to compare the ability of MLPA, NGS and OGM to detect KMT2A-PTDs in relation to the varying detection approaches and gene coverage of KMT2A. Our findings confirm other observations on the complex nature of KMT2A-PTD with respect to KMT2A-PTD size, copy number and location. KMT2A-PTD have been assessed separately using NGS or other methodologies on smaller cohorts (i.e. less than 20) of patients. The study conducted herein is distinct from others by the comparative analysis conducted on 41 AML cases with KMT2A-PTD. This study highlights the complementary aspect of OGM and NGS, two technologies that are used for comprehensive genomic profiling in AML. Several practical questions on KMT2A-PTD remain even following assessment using OGM and NGS. These questions must be addressed before proceeding to a thorough evaluation of the clinical impact of KMT2A-PTD. As mentioned in the discussion, see lines 481-482, we aim to further characterize KMT2A-PTD “OGM and NGS are complementary approaches that can be used to better characterize and interpret KMT2A-PTD in a clinical setting. Several questions on the clinical consequences of the size and composition of KMT2A-PTD are left unanswered and need to be addressed on larger cohorts and likely by additional long-range sequencing characterization and with functional assays.” Several factors prevented the completion of a clinical correlation of KMT2A-PTD with AML outcomes in this study including (1) limited understanding of KMT2A-PTD features (e.g. characterization and interpretation of non-canonical KMT2A-PTD), (2) small KMT2A-PTD study cohort (i.e. 41 patients), (3) short follow-up period (i.e. less than 2 years for 2/3 of the study) and (4) homogeneity of sociodemographic parameters (e.g. 2/3 are AML male patients with AML onset> 60 years).
Is PTD linked to specific oncogenic mutations ?
Studies (e.g. Hinai et al. 2019, PMID: 31723820, Dai et al. 2021. PMID: 34384895, Kim et al. 2019, PMID: 3084064, Vetro et al.2019. PMID: 31698332.) have shown that AML patients KMT2A-PTD harbor additional co-occurring myeloid variants particularly in other epigenetic modifiers (e.g. DNMT3A, IDH1, IDH2, TET2), transcription factors (e.g. RUNX1) and FLT3 internal tandem duplications. Our findings in supplementary Figure 2 (i.e. Figure S2) are in keeping with these observations.
Is there higher genomic instability for these patients ?
A higher complexity of KMT2A-PTD (e.g. higher copy numbers or multiple rearrangement within PTD region) has been described in MDS or AML cases and is believed to be associated with a with an adverse disease outcome (Tsai et al. 2022, PMID: 35584376).
Our study also shows a variety of karyotype results from KMT2A-PTD cases (Table 1). As can be seen in Table 1 normal karyotypes to very complex karyotypes can be seen associated with KMT2A-PTD presence. We would be hesitant to draw conclusions about the effect on genome stability that the KMT2A-PTD without having a larger, matched comparison group. Again, it should be stressed that we believe these are all interesting questions but beyond the scope of this manuscript to address. Our primary goal was to demonstrate the strengths and limitations of NGS, MLPA and OGM for the detection of KMT2A-PTDs including the variability of size and composition that is not detected by NGS and MLPA methods. Any study that will effectively address the reviews concerns would need to be designed from the outset to control for clinical and genomic variables and be sufficiently powered to answer these questions.
Round 2
Reviewer 1 Report
Comments and Suggestions for Authors
I suggest that this manuscript should be accepted in present form.
Reviewer 2 Report
Comments and Suggestions for Authors
I have no more concerns.
Comments on the Quality of English LanguageMinor editing of English language required.